# Further Improvement Based on Traditional Nanocapsule Preparation Methods: A Review

**DOI:** 10.3390/nano13243125

**Published:** 2023-12-12

**Authors:** Yihong Zhou, Peng Wang, Faling Wan, Lifang Zhu, Zongde Wang, Guorong Fan, Peng Wang, Hai Luo, Shengliang Liao, Yuling Yang, Shangxing Chen, Ji Zhang

**Affiliations:** National Forestry and Grassland Bureau Woody Spice (East China) Engineering Technology Research Center, The Institute of Plant Natural Products and Forest Products Chemical Engineering, College of Forestry, Jiangxi Agricultural University, Nanchang 330045, China; zhouyihong@stu.jxau.edu.cn (Y.Z.); wangpeng111@stu.jxau.edu.cn (P.W.); 15970665350@stu.jxau.edu.cn (F.W.); zlf1103@stu.jxau.edu.cn (L.Z.); zongdewang@jxau.edu.cn (Z.W.); jxfgr008@jxau.edu.cn (G.F.); pengwang@jxau.edu.cn (P.W.); luohai@jxau.edu.cn (H.L.); liaosl@jxau.edu.cn (S.L.); wxipyyl@jxau.edu.cn (Y.Y.)

**Keywords:** nanocapsule, preparation, recent progress, physicochemical method

## Abstract

Nanocapsule preparation technology, as an emerging technology with great development prospects, has uniqueness and superiority in various industries. In this paper, the preparation technology of nanocapsules was systematically divided into three categories: physical methods, chemical methods, and physicochemical methods. The technological innovation of different methods in recent years was reviewed, and the mechanisms of nanocapsules prepared via emulsion polymerization, interface polymerization, layer-by-layer self-assembly technology, nanoprecipitation, supercritical fluid, and nano spray drying was summarized in detail. Different from previous reviews, the renewal iteration of core–shell structural materials was highlighted, and relevant illustrations of their representative and latest research results were reviewed. With the continuous progress of nanocapsule technology, especially the continuous development of new wall materials and catalysts, new preparation technology, and new production equipment, nanocapsule technology will be used more widely in medicine, food, cosmetics, pesticides, petroleum products, and many other fields.

## 1. Introduction

Microencapsulation refers to using natural or synthetic polymer encapsulation materials to encapsulate solid, liquid, or even gas core materials, forming a microcapsule with a 1~1000 μm diameter with semi-permeable or sealed membranes [1]. With microencapsulation development, the size of microcapsules can be up to 1~1000 nm [2]. People usually call such microcapsules in the nanometer range nanocapsules. As a multi-phase functional material, nanocapsules have small particle sizes and are easy to disperse and suspend in water to form a clear and transparent colloidal solution [3]. Nanocapsules not only retain the advantages of microcapsules but also make up for the disadvantages of microcapsules [4]. As the particle size reaches the nanometer scale, nanocapsules would have a series of unique phenomena, such as surface, volume, quantum size, and macroscopic quantum tunneling effects. Therefore, nanocapsules have unique properties that are different from traditional microcapsules [5]. For example, nanocapsules can enter cells, tissues, and organs through biofilms [6]; enter blood orally or through injection; and even directly penetrate the skin [7]. Currently, various research results are constantly reported about nanocapsules, and their application scope is gradually being expanded to many fields, such as medicine, food, cosmetics, pesticides, and petrochemicals [8].

Nanocapsule technology mainly refers to using nanocomposites, nanoemulsification, and nanostructure technologies to encapsulate the bioactive substances in the space within the nanometer scale and release the functional components under the action of specific environmental initiators [9]. In addition to the difference in particle size, nanocapsules and ordinary microcapsules are also different regarding wall materials, dispersibility, and preparation methods [10]. This article introduces the preparation techniques of nanocapsules commonly used in chemical, physical–chemical, and physical methods. In addition, it systematically [11] summarizes their preparation principles, advantages and disadvantages, application examples, and the latest research progress.

## 2. Chemical Methods

One of the main chemical methods used to prepare nanocapsules is chemical reactions, such as polymerization and curing reactions, to form the wall materials, which can encapsulate the core material [12]. When preparing nanocapsules using chemical methods, they can be directly crosslinked and cured by chemical reactions without adding a curing agent. The reaction mechanism is clear, and there are many controllable factors in the reaction process. Therefore, chemical methods are widely used to prepare nanocapsules [13]. 

### 2.1. Emulsion Polymerization

The emulsion polymerization method for preparing nanocapsules is based on modifying traditional emulsion polymerization, where monomers are dispersed in water to form an emulsion with the aid of an emulsifier and mechanical stirring. Then, an initiator is added to trigger the polymerization of the monomers. The principle of emulsion polymerization in preparing nanocapsules is that the core material, surfactant, emulsifier, and polymer monomer are dispersed as tiny droplets to form an emulsion at the nanoscale [14]. The polymer monomer can also be present in the mobile phase, which then triggers a polymerization reaction to produce a polymer to coat the core material to make nanocapsules. Depending on the thermodynamic stability of the emulsion, emulsion polymerization for nanomicrocapsules can be divided into miniemulsion and microemulsion polymerizations [15]. 

#### 2.1.1. Miniemulsion Polymerization

Miniemulsion refers to the dispersed stable tiny droplets with a size between 50–1000 nm formed under the action of a high shear force. The droplets contain monomers, dispersed mediums, emulsifiers, co-emulsifiers, initiators, and other components. The polymerization reaction mainly occurs in fine droplets and is called miniemulsion polymerization [16]. Miniemulsion is a thermodynamic metastable system and cannot be formed spontaneously. It must rely on the combined action of a high shear force, emulsifier, and co-emulsifier to overcome the cohesive energy of the disperse phase and the surface energy of forming droplets. So, tiny droplets will be dispersed in the dispersed medium to form miniemulsion [17]. The miniemulsion polymerization is different from the traditional emulsion polymerization. Monomer droplets, rather than micelles, are the main places for polymerization. Therefore, miniemulsion polymerization is more suitable for preparing nanocapsules [18]. A schematic diagram of the preparation of nanocapsules using miniemulsion polymerization is shown in Figure 1. Under the action of emulsifiers and co-emulsifiers, the insoluble core materials and monomers are dispersed into the solution to form emulsion droplets via mechanical agitation, high-speed homogenization, ultrasonic vibration, and other dispersion techniques. The nanocapsules are formed by initiating polymerization reactions in the droplets [19]. 

Depending on different polymerization methods (such as free radical polymerization, ionic polymerization, catalytic polymerization, etc.), different types of monomers can be polymerized in miniemulsion polymerization [20]. Free radical polymerization usually uses vinyl monomers containing unsaturated bonds (such as styrene, acrylate, acrylamide, etc.) as the polymerization raw materials [21]. Free radical polymerization is not limited to the homopolymerization of a single monomer. The copolymerization of two hydrophobic monomers can obtain copolymers with uniform particle sizes. The copolymerization of hydrophilic and hydrophobic monomers can obtain amphiphilic polymers [22]. In ionic polymerization, anionic polymerization can be carried out in aqueous or non-aqueous miniemulsion, such as with cyanoacrylate compounds [23]. Catalytic polymerization is widely used in miniemulsion polymerization, such as the copolymerization of terminal olefins and polyketones [24] and the homopolymerization of terminal olefins [25]. 

Miniemulsion polymerization retains the advantages of conventional emulsion polymerization, such as a high polymerization speed, high molecular weight, good heat dissipation, and low viscosity. In miniemulsion polymerization, the product particle size can be controlled by process parameters, and it also has the characteristics of high stability, a moderate polymerization rate, and so on [26]. The miniemulsion polymerization, which is easy to control during the production process, is very suitable for preparing nanocapsules with a liquid core [27].

Hecht et al. [28] showed that the emulsifier type and amount greatly affected the polymerization of latex particles. The droplet size distribution during emulsification directly affected the properties of nanocapsules. The initiator type also had a significant effect on the miniemulsion polymerization process. The initiator could be divided into two types: oil-soluble and water-soluble initiators. The former can be decomposed in the oil phase, and the latter can be decomposed in the water phase to generate free radicals. The commonly used oil-soluble initiators are benzoyl peroxide, azodiisobutyronitrile, etc. The commonly used water-soluble initiators are potassium sulfate, ammonium persulfate, etc. Shirin-Abadi et al. [29] used n-hexadecane as the core material and studied the effects of the initiator type, polar copolymerization monomer (methacrylic acid), and nonpolar copolymer monomer (2-ethylhexyl acrylate) on the properties of nanocapsules prepared via in situ miniemulsion polymerization. The research showed that different initiators and monomer polarities could affect the morphology and thermal properties of nanocapsules. With the polarity increase in the shell material, the morphology of the encapsulated nanocapsules changed from an ideal round core–shell shape to a half moon shape. The monomer is the basic unit for forming a polymer. Its hydrophilicity, hydrophobicity, concentration, and other characteristics leads to the formation of polymers with different properties. It is very important to select the appropriate monomer in miniemulsion polymerization. Sari et al. [30] prepared nanocapsules of capric, lauric, and myristic acids using miniemulsion polymerization. The research showed that the prepared nanocapsules presented a regular and smooth spherical shape. The average diameter of nanocapsules increased with increases in the fatty acid chain length. The polymerizable emulsifier is a kind of substance that can not only participate in the polymerization reaction but also play the role of an ordinary emulsifier. The traditional emulsifier has poor stability and is easily physically adsorbed on the polymer particles, affecting the film-forming speed and performance of the polymer. Using polymerizable emulsifiers can avoid these problems in miniemulsion polymerization. Chen et al. [31] used polymerizable emulsifier DNS-86 and co-emulsifier hexadecane as raw materials to prepare spherical nanocapsules with an average particle size of 150 nm using a one-step method and miniemulsion polymerization. The research showed that polymerizable emulsifiers played a vital role in preparing nanocapsules. The polymerizable emulsifier amount significantly influenced the emulsion stability, the size of the polymer particles, the thermal properties, and the morphology of nanocapsules. The co-emulsifier addition was beneficial in improving the encapsulation efficiency of nanocapsules.

#### 2.1.2. Microemulsion Polymerization

The microemulsion is an isotropic and thermodynamically stable system that is spontaneously formed by an emulsifier, co-emulsifier, water phase, and oil phase in an appropriate ratio. The microemulsion appearance is transparent or almost transparent. The particle size is generally uniformly distributed between 5 and 200 nm, and the surface tension is very low [32]. Microemulsion has the following characteristics: (1) it can be formed spontaneously without external energy; (2) it is stable in nature and can be stored for a long time without phase separation; (3) it has good solubilization. It can solubilize more lipophilic substances, improving dispersion and bioavailability [33]. According to different energy sources, the microemulsion preparation method can be divided into high-energy and low-energy emulsification methods [34]. The high-energy emulsification method mainly relies on high external energy to complete the emulsification process. According to different equipment, the high-energy emulsification method can be divided into high-pressure homogeneous, micro-jet, and ultrasonic emulsification methods [35]. The low-energy emulsification method mainly relies on the physical and chemical properties of the components and the internal chemical energy of the system to complete the emulsification process. It has the advantages of a low energy consumption, mild reaction conditions, and so on [36]. The low-energy emulsification method mainly includes phase transformation [37], spontaneous emulsification [38], and membrane emulsification methods [39]. 

According to the difference in the oil–water ratio and microstructure in the system, microemulsion polymerization can be divided into microemulsion polymerization (O/W) [40], inverse microemulsion polymerization (W/O) [41], and bicontinuous microemulsion polymerization [42]. Among them, inverse microemulsion polymerization based on the one-pot method is common for preparing nanocapsules. Specific examples are shown in Table 1. In this way, small and uniform nanoparticles with a high relative molecular mass can be rapidly produced in a stable state. According to different polymerization processes and methods, microemulsion polymerization can be divided into seed microemulsion polymerization [43], Winsor I-like microemulsion polymerization [44], and semi-continuous microemulsion polymerization [45]. (See Table 1).

Microemulsion polymerization has the characteristics of a complex phase change of microemulsion and free radical polymerization. Therefore, the factors affecting the phase transition and free radical polymerization can affect the microemulsion polymerization [49]. Among them, the monomer properties and dosage, the initiator type, and especially the use of emulsifiers greatly influence microemulsion polymerization. The properties and dosage of monomers can affect the formation of the microemulsion phase. In the polymerization process, different monomers can lead to different diffusion and exchange behavior in water due to the difference in water solubility and other characteristics. This ultimately leads to the difference in the particle size and particle size distribution of microemulsion polymers [50]. The initiators used for microemulsion polymerization mainly include potassium persulfate, azobisisobutyronitrile, redox initiator, and so on. In addition, there are other initiation methods, such as photoinitiation. Ishizuka et al. [51] discovered the successful glucose oxidase encapsulation in nanocapsules under visible (blue) light using photoinitiation instead of thermal initiation, which avoided enzyme denaturation due to heating. Different initiators will cause significant differences in polymerization efficiency and site and have a greater impact on the particle size of the nanocapsules [52]. Sarkar et al. [53] studied the effects of the monomeric solubility, monomer concentration, and initiator type on polymerization kinetics. The research showed that the better the water solubility of the monomer, the slower the reaction rate, the larger the particle size, and the lower the density of the nanoparticles. The greater the initiator amount, the smaller the latex particle nucleation. The emulsifier is the most critical component in microemulsion polymerization. The main emulsifiers of traditional microemulsion polymerization are as follows: cationic emulsifiers, such as cetyltrimethylammonium bromide (CTAB); anionic emulsifiers, such as sodium dodecyl sulfate (SDS); and nonionic emulsifiers, such as Span series, Tween series, etc. Co-emulsifiers are mainly small-molecule alcohols, such as pentanol, butanol, hexanol, hexadecanol, etc. [54]. Among them, single-chain anionic emulsifiers often need to be combined with co-emulsifiers, such as SDS and pentanol. Cationic and nonionic emulsifiers are often used alone, and some double-chain anionic emulsifiers can also be used alone [55]. For the traditional emulsifier used in microemulsion polymerization, the related research generally has focused on the appropriate emulsifier/oil/water ratio to achieve quantitative particle size control. The research of Liang et al. [56] showed that the greater the emulsifier amount, the smaller the particle size of microemulsion polymerization. Compared with traditional emulsifiers, the research on reactive emulsifiers has gradually become a hot spot. According to the different ways reactive groups participate in polymerization reactions, reactive emulsifiers can be divided into three categories: the surfactant initiator, surfactant chain transfer agent, and polymerizable emulsifier. The first two affect the polymerization kinetics, and the efficiency of the surfactant initiator is not high [57]. At present, the related research is more focused on polymerizable emulsifiers. Zaragoza-Contreras et al. [58] found that the reactive emulsifier P1 (anilinium dodecyl sulfate) could be used as an emulsifier as well as a polyaniline monomer, which could be copolymerized through oxidative polymerization. 

Microemulsion polymerization has the characteristics of providing nano-reaction sites, the formation of a stable phase, and a stable reaction process. Therefore, it is widely used in preparing nanocapsules. The current research on microemulsion polymerization mainly focuses on the research of high-efficiency reactive emulsifiers and developing a high monomer/emulsifier dosage ratio system [59]. 

### 2.2. Interfacial Polymerization

Interfacial polymerization is also often applied to the preparation of microcapsules, requiring polymer monomers to move toward the interface either by their own or external forces. This triggers a polymerization reaction at the interface of the two phases to produce a polymer-coated core. However, interfacial polymerization at the nanoscale requires more precise manipulation. Two monomers containing double (multiple) functional groups are dissolved separately in two different liquids using interfacial polymerization. Then, the polycondensation reaction will occur at the two-phase interface [60]. A fine needle syringe with capillaries is generally used to obtain nano-sized capsule particles. The core material solution and one monomer solution are added to the syringe, then the syringe needle is placed close to the liquid surface of the other monomer solution, and a high-voltage direct current is applied between the needle and the liquid surface. A high-voltage electric field drives the liquid in the syringe to form a uniformly spherical droplet (the particle size is in the nanometer scale) with a surface charge, which is dropped into the second monomer solution. At this time, the two polymerization monomer kinds move from the inside of the two phases to the interface of the droplets and react rapidly at the interface of the two phases to form polymers. The core material is coated by the polymer to form nanocapsules [61]. A schematic diagram of preparing nanocapsules using interfacial polymerization is shown in Figure 2. 

The interfacial polymerization process is simple, the reaction rate is fast, the encapsulation efficiency is high, the equipment is cheap, and the reaction conditions are mild. The interfacial polymerization method does not have high requirements for the reaction monomer purity, but the reaction monomer must have high activity and be able to undergo polymerization. Emulsifiers or co-solvents do not have to be added to the solvent when preparing nanocapsules using interfacial polymerization, but appropriate additions can prevent the aggregation and agglomeration of nanocapsules during the storage of certain nanocapsules. Furthermore, the main role of emulsifiers and catalysts in polymerization at the oil–water interface is to aid the migration of monomers to the oil–water interface. Thus, using emulsifiers or catalysts, nanocapsules are formed spontaneously. The interfacial polymerization method can prepare nanocapsules using water-soluble or oil-soluble substances as the core material. The polymerization degree of the polymer in the production process of interfacial polymerization is uncontrollable, and the formed wall film has a high permeability, which is unsuitable for coating the core material with high sealing requirements. In addition, there are often unreacted monomers and side reactions in the production process, affecting the encapsulation efficiency [62]. Park et al. [63] prepared phase change material (PCM) nanocapsules embedded with magnetic Fe_3_O_4_ nanoparticles through the interfacial polycondensation reaction between toluene diisocyanate and ethylenediamine. They found that the presence of magnetic Fe_3_O_4_ nanoparticles could increase the thermal conductivity and magnetism of nanocapsules. Shi et al. [64] used paraffin as the core and polymethyl methacrylate (PMMA) as the shell to prepare nanocapsules with an encapsulation efficiency of 52.95% and a 200~400 nm particle size through one-step interfacial polymerization. The nanocapsules were spherical, smooth, compact, and uniform, with good stability and degradation.

The preparation of nanocapsules using the chemical method is suitable for solid, liquid, and gas core materials, which are generally active materials. In addition to the above commonly used chemical methods, there are many other chemical methods for preparing nanocapsules, such as in situ polymerization, [65] suspension polymerization [66], the sol–gel method [67], the orifice method, etc. [68].

All of the reaction monomers and initiators are added into the dispersed or continuous phase using in situ polymerization. The polymerization reaction occurs in the dispersed phase since the monomers are soluble in a single phase, and the formed polymers are insoluble in the whole system. The polymerized monomers first form prepolymers and finally form capsule shells on the surface of core materials. Ultrasonic and micro-emulsification techniques are often required to prepare nanocapsules using in situ polymerization [69]. (See Table 2).

The suspension polymerization method directly uses the polymer as the raw material. The polymer monomer is dissolved in the organic phase, then suspended, crosslinked, solidified, and precipitated on the organic phase surface to form the shell material. The advantages of suspension polymerization are that the reaction is easy to control and the product is easy to separate and handle [73]. 

The sol–gel method mainly uses liquid compounds (such as metal alkoxides) containing highly chemically active components as precursors. The precursors are mixed evenly with solvents, catalysts, complexing agents, etc., under liquid phase conditions, and a stable sol system is formed in the solution through chemical reactions, such as hydrolysis and condensation. After the sol is aged, the colloidal particles further polymerize to form a gel with a three-dimensional network structure. The gel is dried and solidified to form nanocapsules [74]. 

The soluble wall and core materials are mixed using the orifice method then dropped into the coagulation liquid through the orifice. Under the action of the coagulation liquid, the high-molecular polymer solidifies to form nanocapsules. The equipment of the orifice method is simple, and the cost is low. Under low temperatures, nanocapsules with uniform particle sizes can be prepared [75]. 

## 3. Physicochemical Methods

The preparation of nanocapsules using the physicochemical method is mainly performed through changing the temperature, the pH, adding electrolytes, etc. The dissolved film-forming material is precipitated and deposited on the core material surface to form the wall material. The physicochemical method is mainly used for coating solid and liquid core materials, especially those that are volatile and have poor thermal stability [76]. 

### 3.1. Layer-by-Layer (LBL) Self-Assembly

LBL self-assembly is a technology that uses certain affinity characteristics of specific polymer compounds to spontaneously deposit them layer-by-layer alternately on the surface of template materials to form molecular aggregates with a complete structure and stable performance [77]. In the nanocapsule preparation using LBL self-assembly, colloidal particles (such as latex or cells) with active components are usually used as assembly templates to alternately absorb substances with opposite charges, such as polyelectrolytes (including polyanions and polyanions), enzymes, antibodies, viruses, or inorganic nanoparticles. Through electrostatic attraction, they polymerize or deposit on the surface of template particles and self-assemble layer-by-layer to form two-dimensional or three-dimensional nanoparticles with a core–shell structure. The colloidal particles of the template can be removed through dissolution or melting using different templates and wall materials, and various nanocapsules with a 10~40 nm wall thickness can be prepared [78]. A schematic diagram of the preparation of nanocapsules using LBL self-assembly is shown in Figure 3. When the core part of the nanocapsules is removed through dissolution or heating and burning, the solvent dissolution conditions are harsh, and a part of the products will remain inside the microcapsules after the decomposition. When the template is removed through calcination, the hollow nanocapsules need to withstand higher temperatures and the gas pressure generated by the cracking of the template, which may cause damage to the nanocapsules. In recent years, more and more researchers have adopted microemulsion as templates. This template type has a small particle size, uniform size, and mild preparation conditions, and the most important thing is that it does not require a denucleation process [79]. Shabbar et al. [80] prepared a nanoemulsion with a particle size of 142.7 ± 0.85 nm using modified starch. Nanocapsules with an average particle size of 159.85 ± 0.92 nm were obtained via LBL self-assembly using a nanoemulsion as the template and chitosan and sodium carboxymethylcellulose as wall materials. 

Studies have shown that the assembly driving force is the key to realizing LBL self-assembly. The driving force types of LBL nanocapsule assembly include static electricity, hydrogen bonding, a covalent bond, host–guest interactions, and so on [81]: (1)The electrostatic assembly driving force is the most classic and mature type in the LBL self-assembly method. However, the nanocapsules prepared based on static electricity have a weak stability and are easily destroyed under extreme conditions, such as high temperatures, strong acids, strong alkalis, and high ionic strengths [82]. Fan et al. [83] used PLGA as the core and alternately wrapped poly-L-ornithine (PLO) and the sulfated polysaccharide rockweed polysaccharide complex in the outer layer to form a shell, successfully encapsulating the drug dispersed within the PLGA core and allowing a controlled release.(2)The preparation of nanocapsules based on hydrogen bonding interactions is similar to electrostatic interactions, which are all weak interactions. The hydrogen-bonding self-assembly generally needs to suppress the ionization of assembled molecules, so the wall of the prepared nanocapsules is sensitive to pH. Hwangbo et al. [84] used polyethyleneimine and siloxane coupling agents as the wall material, silica as the core material, and hydrogen-bonding self-assembly to successfully prepare nanocapsules with antibacterial functions and good hydrophilicity.(3)Through the LBL self-assembly of covalent bond interactions, a stable covalent bond is formed that connects the layers, and a firm self-assembly structure can be obtained. This is due to the chemical reaction of the functional groups of the assembly material. The prepared nanocapsules have a good stability and are not easy to degrade under strong acids, strong bases, high salinity, high temperatures, and so on [85]. Zhang et al. [86] successfully used covalent LBL self-assembly (CSA) to synthesize hollow capsules with N-Methyl-2-nitro-diphenylamine-4-diazoresin (NDR) and m Methylphenol-formaldehyde resin (MPR) in latex core LBL assembly to form nanocapsules with an average diameter of 260 nm.(4)The host–guest interaction is a common assembly technique in supramolecular chemistry. The LBL microcapsules based on the host–guest effect usually have excellent environmental responsiveness [87]. Li et al. [88] assembled bis-aminated poly(glycerol methacrylate)s and cucurbit [7] uril on mesoporous silica nanoparticles. As a molecular bridge, cucurbit [7] uril connected two different diamino polymer molecular layers through the interaction between the host and guest. The prepared nanocapsules could release the encapsulated anticancer drug (doxorubicin hydrochloride) under specific acidic conditions. (See Table 3).

In addition to the above acting forces, other acting forces can also be used as driving forces for forming LBL nanocapsules, such as charge transfer interaction, chemical crosslinking, specific molecular recognition, halogen bonds, and base pairs. There are differences in the physical and chemical structures, intelligent responsiveness, and stability of nanocapsules prepared with different assembly driving forces, broadening the application range of LBL self-assembly technology and providing technical support for preparing nanocapsules [93]. 

The experimental equipment for preparing nanocapsules through LBL self-assembly is simple, and the preparation conditions are mild. There is no need for organic solvents or strong acids, strong alkalis, high temperatures, and other harsh conditions. It is an environmentally friendly preparation method. The advantage of the LBL method in preparing nanocapsules is that it can accurately control the size, composition, thickness, structure, and surface state of the nanocapsules [94]. For example, the size and shape of the capsule cavity can be controlled through the diameter and shape of the template colloidal particles; the wall thickness and surface characteristics of the nanocapsule can be adjusted through the number of depositions and the polyelectrolyte type; the capsule wall microstructure can be adjusted by changing the charge density of the polyelectrolyte, the rigidity of the molecular chain, and the solution conditions (such as ionic strength, pH), so as to change the affinity and transmittance of the capsule wall [95]. Fan et al. [96] constructed a double-layer nanocapsule encapsulated using LBL self-assembly. The inner layer was a silicon precursor (tetramethoxysilane), and the outer layer was chitosan. The vanillin encapsulation efficiency could be as high as 95.5%. The size of the nanocapsules could be adjusted by changing the chitosan amount. The double-layer structure could form a double barrier to vanillin, effectively delaying the release rate of spices. Since the charge on the nanocapsule surface could be stably dispersed, the surfactant is not needed in the preparation process.

Nanocapsules prepared using LBL self-assembly technology have potential application prospects in the fields of biochemistry, pharmacy, catalysis, and so on. For example, the smart responsive nanocapsules prepared using LBL self-assembly will break and release encapsulated substances when stimulated by temperature, pH, light, the magnetic field, and other external factors. So, they can be widely used in controlled drug release, nanoreactors, sensors, and other fields. However, a small amount of impurities may affect the self-assembly behavior and application performance due to the strict LBL system requirements. Currently, the LBL method is suitable for laboratory research and difficult to apply in industrial production [97].

### 3.2. Nanoprecipitation

Nanoprecipitation is mainly used to prepare nanoparticles by controlling the mixing of a solute solution and nonsolvent. A schematic diagram of nanocapsules prepared using the nanoprecipitation method is shown in Figure 4. The polymer is dissolved in an organic solvent, and the oil-soluble surfactant and the core material are added. The obtained organic solution is added to the aqueous phase containing the water-soluble surfactant under stirring conditions. The polymer can be deposited on the surface of the oil droplets, thus forming a stable colloidal suspension. The organic solvent is removed under reduced pressure to obtain a colloidal solution of nanocapsules [98]. The commonly used organic solvents in nanoprecipitation are ethanol, acetone, hexane, dichloromethane, dioxane, and so on. The nanoprecipitation method is simple and mild. The particle size of the products can be adjusted by changing the reaction conditions.

Nanocapsules with a narrow size distribution can be obtained [99]. However, this method is suitable for coating hydrophobic substances easily soluble in organic solvents, limiting its application in water-soluble substances [100]. Miladi et al. [101] studied the nanoprecipitation process. 

The study found that the key variables of the nanoprecipitation process were the relevant condition variables for adding the organic phase to the aqueous phase, such as the injection rate of the organic phase, the stirring rate of the aqueous phase, the method of adding the organic phase, the ratio of the organic phase to the water phase, and the surfactant properties and concentration. Changes in these parameters will greatly affect the physical properties and encapsulation efficiency of nanocapsules. Badri et al. [76] studied the influence of process and formulation parameters on the polycaprolactone nanocapsule preparation via nanoprecipitation. The research showed that the appropriate increase in the organic phase injection rate would promote a nucleation rate increase and then reduce the average particle size. A faster stirring speed would bring a faster diffusion speed, thereby promoting solvent diffusion. 

The increase in the aqueous phase volume is helpful to the diffusion of water-soluble solvent in the aqueous phase. The excessively fast diffusion speed could even cause the polymer to precipitate immediately before being aggregated into nanoparticles. The polymer and core material concentration would greatly affect the nucleation and growth of nanoparticles and the encapsulation efficiency of the core material.

In recent years, the development of flash nanoprecipitation technology has greatly promoted the practical application of nanoprecipitation technology [102]. This is based on the hedge collision mixer and the multi-entry vortex mixer and the application of advanced automation and high-throughput technologies (such as microfluidic technology and the pipetting robot) in nanoprecipitation [103]. These technologies are applied in the nanoprecipitation method, which can realize the rapid mixing of the reaction system and the convenient adjustment of the preparation parameters, assisting in realizing the automation, scale, and continuity of the preparation of nanocapsules. With the development of materials science and nanoscience, the nanoprecipitation method is expected to be further developed and improved, and its application will be further expanded [104].

There are many ways to prepare nanocapsules using physicochemical methods. In addition to the above-mentioned common methods, they also include coagulated phase separation and multiple emulsion–solvent evaporation methods [105,106]. (See Table 4).

The preparation of nanocapsules through the condensed phase separation method is mainly based on the continuous phase composed of the mixed solution of the core and wall materials. Adding electrolyte inorganic salt, a nonsolvent of the wall material or a coagulant or changing the concentration, temperature, and pH in the continuous phase reduces the solubility of the wall material polymer. The wall material polymer is condensed out of the solution then deposited and coated on the core material surface to form nanocapsules. According to the different dispersion mediums, the condensed phase separation method can be divided into aqueous and oil phase separation methods. The aqueous phase separation method is suitable for coating hydrophobic core materials, while the oil phase separation method is suitable for coating hydrophilic core materials [111]. According to different film-forming materials, the aqueous phase separation method is divided into single and complex coacervation methods. The single coagulation process is simple and has a high encapsulation efficiency. It can encapsulate water-insoluble solids or liquids, such as oils and essential oils. It is difficult to control the particle size of nanocapsules in a single coagulation system, and the cost is relatively high. The complex coagulation method with two or more polymer polyelectrolytes as wall materials is widely used to prepare nanocapsules. It is a high-yield and high-efficiency encapsulation method and very suitable for coating core materials as a water-insoluble solid powder or liquid, especially some unstable substances, such as polyphenols [112]. 

The multiple emulsion–solvent evaporation method mainly disperses the mixed solution of the wall and core materials as droplets into a volatile medium. The volatile dispersion medium is quickly removed from the droplets to form the capsule wall. Then, through certain methods (such as heating, decompression, stirring, and solution extraction), the solvent in the capsule wall is removed to achieve core encapsulation. The key to preparing nanocapsules using the multiple emulsion–solvent evaporation method is to control the size of the emulsion droplets formed by the solvent before evaporation. Nanocapsules can be prepared by changing the stirring rate, the emulsifier type and amount, the emulsification way, adjusting the viscosity and the proportion of the organic and aqueous phases, etc. The disadvantages of this method are that the reaction time is long, the reaction process is difficult to control, and the product yield is low [113].

## 4. Physical Methods

The physical method is the preparation method of nanocapsules using physical and mechanical principles. The physical method has the characteristics of simple equipment, a low cost, being easy to popularize, and large-scale continuous production [114].

### 4.1. Supercritical Fluid (SCF)

SCF refers to a high-density liquid at or above the critical temperature and pressure values. Common supercritical fluids include ethane, ethylene, propane, propylene, ammonia, water, carbon dioxide, etc. Among them, supercritical carbon dioxide fluid is the most widely used. SCF technology uses the special physical properties of supercritical fluids and the different solubilities of solutes and solvents in supercritical fluids to prepare nanocapsules. This method can reduce the content of harmful residual components in the nanocapsules and eliminate the organic solvent or minimize its use, and it is particularly suitable for processing heat-sensitive substances. The prepared nanocapsules have a small particle size and a narrow distribution range [115]. 

There are three main methods for preparing nanocapsules using a SCF: the rapid expansion of the supercritical fluid (RESF) [116], a supercritical antisolvent (SAS) [117], and the supercritical fluid extraction of emulsions (SFEE) [118].

#### 4.1.1. Rapid Expansion of the Supercritical Fluid

The rapid expansion of the supercritical fluid method for preparing nanocapsules is based on the principle of fluid volume changes caused by pressure changes near the critical point. This method is mainly applied to substances with good solubility in supercritical fluids. Through the quick pressure release, the substances quickly reach supersaturation and precipitate to form fine particles. The rapid expansion of the supercritical fluid is a relatively mature method of supercritical fluid technologies [119]. Ladawan et al. [120] used the RESF to encapsulate tetrahydro curcumin (THC) into poly(L-lactide) (PLLA). The pressurized ethanol and carbon dioxide mixture was used as the solvent, and the solute was precipitated to form particles by rapidly expanding the feed liquid into the water. The prepared products were all spherical, and the THC had a high drug loading. After the rapid expansion process, the antioxidant activity of THC remained unchanged, and the encapsulation in PLLA also had a slow-release effect. However, the controllability of this method was not high, the preparation result was not stable, and the wall and core materials were required to be dissolved in supercritical carbon dioxide, limiting the application range of this method.

#### 4.1.2. Supercritical Antisolvent (SAS) 

SAS technology can be divided into the following types.

Gas antisolvent (GAS)

In the GAS method, a supercritical antisolvent is added to the solution to expand the solvent in the solution, thus reducing the solute solubility in the solution. In this process, the supercritical fluid injection rate is the main factor affecting the particle size and shape [121]. 

Aerosol solvent extraction (ASE)

The principle of this method is similar to the GAS method. The difference is that the solution is injected into the antisolvent in the ASE method, making the solution reach saturation quickly. The precipitated particles are more uniform, and the drying effect is better. ASE is one of the most commonly used methods to prepare nanoparticles in a supercritical fluid, and many derivatization methods have also been developed based on it [122]. 

Solution-enhanced dispersion (SED)

In the SED method, the supercritical fluid and solution are simultaneously injected into the precipitator through a coaxial nozzle to enhance the mass transfer effect to achieve a better atomization effect and precipitate out finer particles [123]. 

Solution-enhanced dispersion via ultrasound (SED-U)

The SEDS-U process is based on adding ultrasonic assistance to the SEDS method. This method can strengthen the mass transfer effect of the solution and the antisolvent, control the particle size distribution, and make the particle size smaller [124]. A schematic diagram of the preparation of nanocapsules using the SED-U method is shown in Figure 5. SAS technology is one of the most promising supercritical technologies. The nanoparticles prepared using the SAS method have the advantages of a narrow particle size distribution, less organic solvent residue, simple operation, and environmental protection. Campardelli et al. [125] successfully used poly-lactic-co-glycolic (poly-lactic-co-glycolic) as the wall material to encapsulate different nonsteroidal anti-inflammatory drugs using the improved SAS method. The prepared nanocapsules had a concentrated particle size distribution and a good sustained-release effect. The encapsulation efficiency of the core material was between 50% and 97%. The critical SAS technology parameter is the encapsulation efficiency, which depends on many factors, especially the initial concentration of the wall and core materials. This method is more suitable for nanoencapsulation with oil-soluble substances as the core materials, limiting its application [126]. 

#### 4.1.3. Supercritical Fluid Extraction of Emulsions (SFEE) 

The SFEE method is a new method of preparing nanocapsules based on SAS technology and the emulsion method. This method uses O/W emulsion or W/O/W multiple emulsion as raw materials and dissolves the core material in the oil phase or inner water phase of the emulsion. Then, the supercritical fluid is used to extract the organic solvent of the emulsion, and the wall and core materials are precipitated to form a precipitate [127]. The research by Lévai et al. [128] showed that the size of the nanocapsules mainly depended on the size of the emulsion droplets. A high encapsulation efficiency and good dispersibility, the stability, the droplet size, and the uniformity of the raw material emulsion should be ensured first to ensure that the prepared nanocapsules will have small and uniform particle sizes. The requirement of this method for the core material is only that it be soluble in the oil phase or the inner water phase, greatly broadening the application range of the supercritical method for preparing nanocapsules. (See Table 5).

### 4.2. Nano Spray Drying

Spray drying uses an atomizer to atomize the feed liquid into tiny droplets. It rapidly evaporates water and other solvents in the feed liquid during direct contact with hot air or another hot medium in the drying tower. After the liquid material is dried into powder products in a very short time, they are discharged from the drying tower to the cyclone separation tower to realize gas–solid separation and obtain the dried products. However, in a traditional spray drying experimental device, the typical cyclone separator cannot collect particles below 2 µm. Even if a high-performance glass cyclone is used, the particle size cannot be reduced below 1.4 µm. A feasible method for collecting nanoparticles is to use an electrostatic particle collector. In addition, the traditional atomizer cannot produce very fine droplets to dry solid particles in the sub-micron range. It is necessary to transform conventional spray drying equipment to produce nanocapsules using spray drying technology [4]. 

The Nano Spray Dryer B-90 launched by Büchi Labortechnik AG of Switzerland in 2009 enabled the preparation of nanocapsules using the spray drying method. The nano spray dryer comprised a high-frequency vibration atomizer, laminar flow heater, and high-voltage electrostatic collector [131]. A schematic diagram of the nano spray drying process is shown in Figure 6, which was mainly divided into the following three basic steps: (1) The feed liquid entered the high-frequency vibration atomizer. The porous metal diaphragm in the atomizing spray head was driven via high-frequency ultrasound to vibrate up and down, producing millions of precise and fine droplets per second. (2) The droplets entered the dry gas. The dry gas was heated via laminar heating through a compact porous metal foam. The droplets were contacted with the drying medium, and the solvent was evaporated to produce dry nanoparticles. (3) The dried nanoparticles were charged with static electricity, and the nanoparticles were efficiently separated from the dry air through the electrostatic particle collection electrode. Finally, the drying gas was removed from the spray dryer and then filtered by a filter [132]. 

Nano spray drying technology has the following advantages in preparing nanocapsules: a mild heating process; a simple production process; fast drying rate; a high yield (up to 90%); uniform particle size distribution and a wide distribution range; dry particles have good dispersibility; and it is very suitable for processing heat-sensitive materials. Therefore, it has broad application prospects in the fields of pharmaceutical preparations, cosmetics, chemistry, food, etc. In the nano spray drying process, the spray net size, spray rate, feed liquid composition and concentration, inlet temperature, and dry gas flow rate will significantly affect the size, morphology, dispersity, and stability of the nanocapsules [133]. Common examples of nano spray drying for preparing nanocapsules are shown in Table 2. Among them, Nguyen et al. [134] studied the effect of the chitosan molecular weight and spray dryer nozzle size on the average particle size and zeta potential of chitosan nanoparticles. The results showed that when the molecular weights of chitosan were 130, 760, and 1200 cPs, the average particle sizes of chitosan nanoparticles were 166.7, 334.6, and 1230.0 nm, respectively. When the nozzle size increased from 4.0 to 7.0 µm, the average size of the nanoparticles increased from 156.1 to 376.1 nm, and the zeta potential value decreased from 59.3 to 50.0 mV. The smaller the nozzle size, the lower the molecular weight of chitosan, and the higher the zeta potential, the smaller the size of chitosan nanoparticles. Hu et al. studied the eugenol nanocapsules prepared using the nano spray drying method through scanning electron microscopy (SEM). The study found that when the nozzle size diameter was 4 μm, most prepared nanocapsules were smooth and flat spheres. With increases in the nozzle size diameter and the liquid material concentration, the number of folded and hollow spheres in the particles increased, and the particle size of the nanocapsules also increased. Öztürk et al. [135] studied the effect of the inlet temperature, nozzle size, and feed liquid concentration on the properties of the prepared nanocapsules. The results showed that the inlet temperature mainly affected the yield and the encapsulation efficiency. However, the nozzle size mainly affected the particle size. The feed liquid concentration mainly affected the particle volume and diameter. The research of Amsalem et al. [136] showed that using organic volatile solvents (such as acetone, dichloromethane, and acetonitrile) could achieve an effective nano spray drying process at low temperatures (<60 °C). This technology was particularly suitable for drying heat-sensitive biological macromolecules, such as siRNA and proteins. The physical and chemical properties of nanocapsules were mainly affected by various parameters of the preparation process and nano spray drying equipment. (See Table 6).

## 5. Conclusions

Nanocapsule technology, as an interdisciplinary subject, involves various technologies, such as physics and colloid chemistry, polymer physics, and chemistry, dispersion and drying technology, nanomaterials, and nanoprocessing in nanotechnology. There are various methods for preparing nanocapsules. To obtain nano-sized particles, the preparation method of nanocapsules is more complicated than that of ordinary microcapsules. Selecting the appropriate method to prepare nanocapsules under the given conditions requires careful consideration of the properties of the core and wall materials, the particle size of the nanocapsule product, the release mechanism of the core material, and the cost. The future development trend of nanocapsules as a composite phase functional material will be in the direction of a small particle size, narrow particle size distribution, good dispersion, high selectivity, wide application range, and strong environmental adaptability.

At present, the nanocapsule technology has achieved many research results. However, there are many deficiencies in both theory and application, and more in-depth research is needed. For example, many existing nanocapsule preparation methods, such as emulsion polymerization and supercritical fluid technology, still have the following problems. The encapsulation efficiency of the core material is low; the size, distribution, uniformity, and roundness of the formed nanocapsules cannot be controlled; the functionality of the nanocapsules is difficult to predict accurately; and the stability of the nanostructure and properties of the capsules is not high. In addition, the preparation cost of nanocapsules is generally high today, and the industrialization development is not very satisfactory. Many nanocapsule technology projects are still only at the laboratory level, and they cannot achieve the smooth transformation of nanocapsule technology achievements from basic research to industrialization. With the deepening of the research and understanding of nanocapsule technology, especially the continuous development of new wall materials, new preparation technology, and new production equipment, the huge advantages brought by nanocapsule technology will also play a positive role in promoting its development, and its application fields will be more extensive.

## Figures and Tables

**Figure 1 nanomaterials-13-03125-f001:**
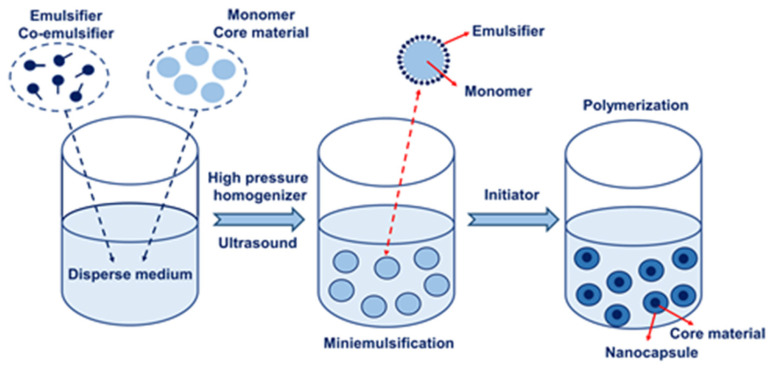
A schematic diagram of the preparation of nanocapsules via miniemulsion polymerization.

**Figure 2 nanomaterials-13-03125-f002:**
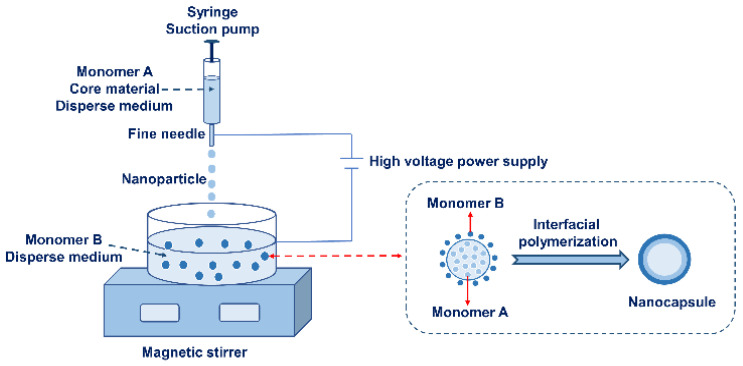
A schematic diagram of the preparation of nanocapsules using interfacial polymerization.

**Figure 3 nanomaterials-13-03125-f003:**
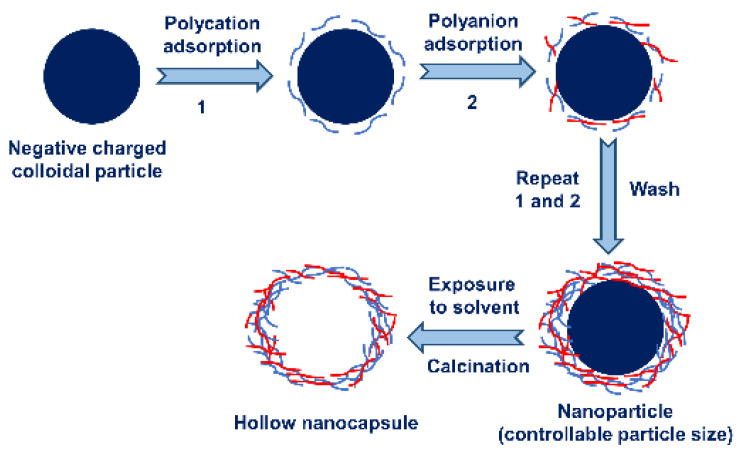
A schematic diagram of the preparation of nanocapsules using layer-by-layer self-assembly.

**Figure 4 nanomaterials-13-03125-f004:**
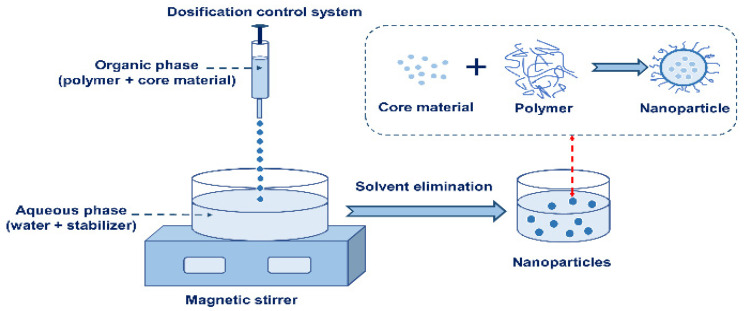
A schematic diagram of the preparation of nanocapsules using nanoprecipitation.

**Figure 5 nanomaterials-13-03125-f005:**
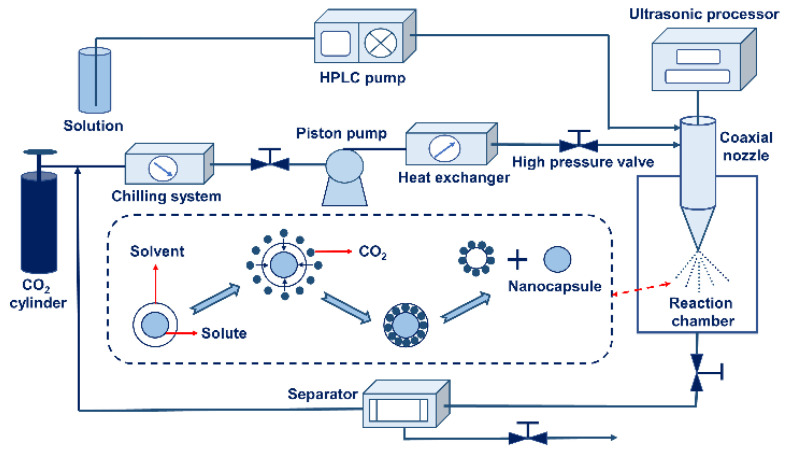
A schematic diagram of the preparation of nanocapsules using SED-U.

**Figure 6 nanomaterials-13-03125-f006:**
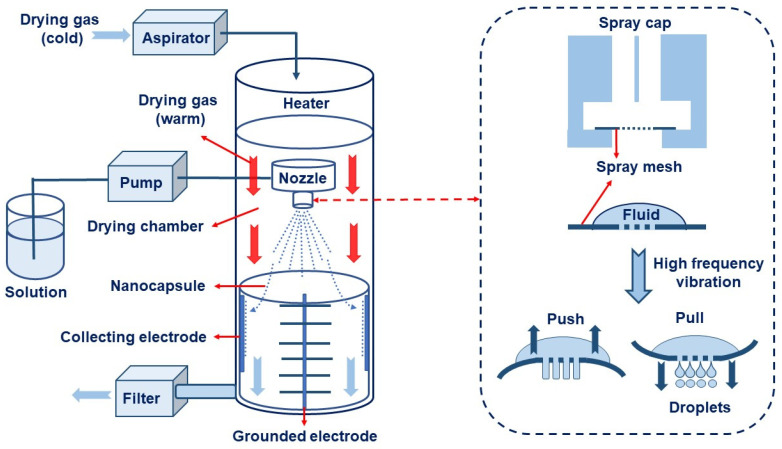
A schematic diagram of the preparation of nanocapsules using nano spray drying.

**Table 1 nanomaterials-13-03125-t001:** Preparation of nanocapsules via inverse microemulsion polymerization.

Core	Monomer	Emulsifier	Encapsulation Efficiency (%)	Particle Size (nm)	References
Caffeine	Whey protein, sugar beet pectin	Sorbitan monooleate (Span-80)	83	200	Gazme et al. [46]
Develop Docetaxel	Polycaprolactone	Tween 80, Span 80	65	180~210	Daşkın et al. [47]
UV filters	Methyl methacrylate	Styrene-co-methacrylic acid	69.85	50-500	Chen et al. [48]
Ivermectin	Poly(ε-caprolactone) (PCL)	Span 80 and Tween 20	98~100	400	Souza et al. [11]

**Table 2 nanomaterials-13-03125-t002:** Preparation of nanocapsules using interfacial polymerization.

Wall Material	Solvent	Core Material	Reaction Temperature (°C)	Particle Size (nm)	TSM or SEM	References
Polymethyl methacrylate	Water	Paraffin	80	200~400	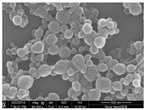	Shi et al. [64]
Polypyrrole	Water		0	200~500	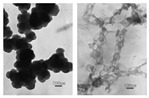	Zhang et al. [70]
Arginine polyamide	Water and acetone	Promethazine hydrochloride	0~5	193.63	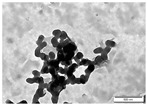	Alyami et al. [71]
Chitosan and poly(N-vinyl pyrrolidone-alt-itaconic anhydride)	Water and acetone	Span 80 and Tween 20	65	107~250	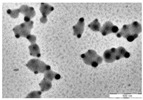	Dellali et al. [72]

**Table 3 nanomaterials-13-03125-t003:** Preparation of nanocapsules via layer-by-layer self-assembly.

Core Material or Template	Monomer 1	Monomer 2	Particle Size (nm)	TSM or SEM	Refs.
Curcumin-enriched MCT oil nanoemulsion	Chitosan	Allylamine hydrochloride	159.85	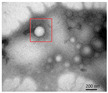	Shabbar et al. [80]
Silica nanoparticles	Branched polyethyleneimine	Silsesquioxane	Approximately 58.83	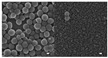	Hwangbo et al. [84]
Resveratrol nanoparticles	Allylamine hydrochloride	Anionic dextran sulfate	116~220	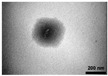	Santos. Et al. [89]
Poly(lactide-coglycolide) (PLGA)	L-ornithine	Sulfated polysaccharide fucoidan	170	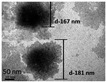	Fan et al. [83]
2-Ethylhexyl-4-dimethylaminobenzoate	Chitosan	Sodium alginate and calcium	155~205	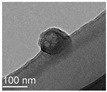	Xu et al. [90]
Soybean oil	Egg lecithin	Glycol chitosan	106~130	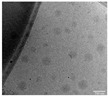	Vecchione et al. [91]
Cerium oxide nanoparticles and pirfenidone	Poly-1-arginine (parg)	Dextrose sulfate (DS)	111.1	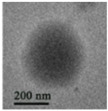	He et al. [92]

**Table 4 nanomaterials-13-03125-t004:** Preparation of nanocapsules using nanoprecipitation.

Wall Material	Core Material	Organic Solvent	Surfactant	Particle Size (nm)	Encapsulation Efficiency (%)	Refs.
PCL	Perillyl alcohol and chitosan	Acetone	Sorbitan monoestearate	330	56	Penteado et al. [107]
Polycaprolactone	Quercetin	Acetone	Tween 80 and Span 80	227.8	92.5	Mahmoud et al. [108]
Pcl	Icaridin	Acetone	Tween 80	314	98.7	Andrade et al. [109]
PLC and sorbitan monostearate	Essential oil	Acetone	Tween 80	210	93	Granata et al. [110]

**Table 5 nanomaterials-13-03125-t005:** Preparation of nanocapsules using supercritical fluid.

Wall Material	Core Material	Pressure	Reaction Temperature (°C)	Particle Size (nm)	Encapsulation Efficiency	Method	TSM or SEM	Refs.
Polycaprolactone	Vitamin E	8.0	40	9	Above 70%	SFEE	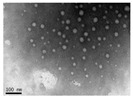	Prieto et al. [115]
Hydroxypropylmethyl cellulose phthalate	Lutein	11	40	163 ~ 219	88.41%	SAS	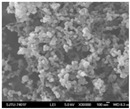	Jin et al. [129]
Multiple polymer monomers	The dye (Nile Blue)	5	25	40~100		RESS	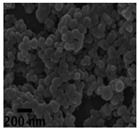	Dong et al. [130]
Polycaprolactone	Quercetin	9	40	25~35	82.4%	SFEE	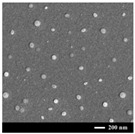	Dong et al. [126]

**Table 6 nanomaterials-13-03125-t006:** Preparation of nanoparticles using nano spray drying.

Wall Materials	Core Materials	Solvent	Inlet Temperature (°C)	Drying Gas (L/min)	Particle Size (nm)	Encapsulation Efficiency (%)	Product Yield (%)	SEM	References
Chitosan, tripolyphosphate	Amoxicillin trihydrate	0.35% (*w*/*v*) acetic acid solution	120	120	156~376	——	90	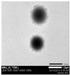	Nguyen et al. [134]
Hyaluronic acid, poly(acrylic acid)	Acyclovir	Water	85	——	258	85	75	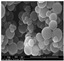	Sithole et al. [137]
Kollidon^®^ SR,Eudragit^®^ RS	Dexketoprofen trometamol	Methanol	119~121	——	108~691	35~51	——	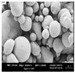	Öztürk et al. [135]
Poly (D,L-lactide-co-glycolide acid)	Human serum albumin primary nanoparticles loaded with sirna	Acetonitrile	50	118~121(N_2_/CO_2_)	270~990	20~25	60	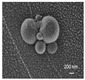	Amsalem et al. [136]
Whey protein isolate	Roasted coffee bean oil	Water	90	90~110	206~404	——	——	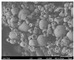	Prasad Reddy et al. [138]
Gum Arabic, lecithin	Eugenol	Water,Ethanol	100	100~110	317~491	——	——	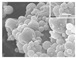	Hu et al. [139]
β-cyclodextrin	Hydroxytyrosol	Water	100	100	400~3400	81~88	53	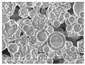	Malapert et al. [140]
Gum arabic and maltodextrin	Pistacia terebinthus fruit oil	Water	135	500		93.33	45.27	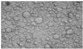	Yaman et al. [141]
Protein isolate-maltodextrin mixtures	Oregano essential oil	Water	100	130		77.9	54.9	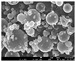	Plati et al. [142]

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
