# Peer review of "Further Improvement Based on Traditional Nanocapsule Preparation Methods: A Review"

_nanomaterials, 2023, doi:10.3390/nano13243125_

Round 1
Reviewer 1 Report
Comments and Suggestions for Authors
Dear, the review does not provide a complete and clear overview of nanocapsules synthesis protocols and applications. Works on microemulsions, nanoemulsions, nanoparticles (nanoprecipitation) are presented and discussed. Many references are outdated.
It would be more appropriate to restructure the entire work with a view to discussing nanoencapsulation methods rather than the synthesis of nanocapsules as presented by the authors.
Comments on the Quality of English LanguageThe English language and syntax need a complete overhaul. Some sentences are far removed from scientific language and make reading difficult.
Reviewer 2 Report
Comments and Suggestions for Authors
Dear Authors,
The overview is clear, comprehensive and relevant to the field. There are many recent reviews on encapsulation methods, but the current review is still relevant and of interest to the scientific community. However, it is recommended to increase the number of citations of the most recent in exchange for older items. The most recent are not in the vast majority. Figures are adequate, although not very eye-catching. It is recommended to enrich the tables with more recent examples.
Round 2
Reviewer 1 Report
Comments and Suggestions for Authors
I consider the authors' revision work sufficient for publication.